# Repurposing of High-Dose Erythropoietin as a Potential Drug Attenuates Sepsis in Preconditioning Renal Injury

**DOI:** 10.3390/cells10113133

**Published:** 2021-11-12

**Authors:** Wiwat Chancharoenthana, Kanyarat Udompronpitak, Yolradee Manochantr, Piyawat Kantagowit, Ponthakorn Kaewkanha, Jiraporn Issara-Amphorn, Asada Leelahavanichkul

**Affiliations:** 1Tropical Nephrology Research Unit, Department of Clinical Tropical Medicine, Faculty of Tropical Medicine, Mahidol University, Bangkok 10400, Thailand; wiwat.cha@mahidol.ac.th; 2Department of Microbiology, Faculty of Medicine, Chulalongkorn University, Bangkok 10330, Thailand; jubjiibb@hotmail.com (K.U.); e4rn.yola@gmail.com (Y.M.); kantpiya@windowslive.com (P.K.); first@docchula.com (P.K.); jiraphorn298@gmail.com (J.I.-A.); 3Translational Research in Inflammation and Immunology Research Unit (TRITU), Department of Microbiology, Chulalongkorn University, Bangkok 10330, Thailand

**Keywords:** cecal ligation and puncture, erythropoietin, LPS, macrophages, mortality rate, sepsis

## Abstract

Due to (i) the uremia-enhanced sepsis severity, (ii) the high prevalence of sepsis with pre-existing renal injury and (iii) the non-erythropoiesis immunomodulation of erythropoietin (EPO), EPO was tested in sepsis with pre-existing renal injury models with the retrospective exploration in patients. Then, EPO was subcutaneously administered in mice with (i) cecal ligation and puncture (CLP) after renal injury including 5/6 nephrectomy (5/6Nx-CLP) and bilateral nephrectomy (BiNx-CLP) or sham surgery (sham-CLP) and (ii) lipopolysaccharide (LPS) injection, along with testing in macrophages. In patients, the data of EPO administration and the disease characteristics in patients with sepsis-induced acute kidney injury (sepsis-AKI) were evaluated. As such, increased endogenous EPO was demonstrated in all sepsis models, including BiNx-CLP despite the reduced liver erythropoietin receptor (EPOR), using Western blot analysis and gene expression, in liver (partly through hepatocyte apoptosis). A high-dose EPO, but not a low-dose, attenuated sepsis in mouse models as determined by mortality and serum inflammatory cytokines. Furthermore, EPO attenuated inflammatory responses in LPS-activated macrophages as determined by supernatant cytokines and the expression of several inflammatory genes (*iNOS*, *IL-1**β*, *STAT3* and *NF**κ**B*). In parallel, patients with sepsis-AKI who were treated with the high-dose EPO showed favorable outcomes, particularly the 29-day mortality rate. In conclusion, high-dose EPO attenuated sepsis with preconditioning renal injury in mice possibly through the macrophage anti-inflammatory effect, which might be beneficial in some patients.

## 1. Introduction

Sepsis is a life-threatening condition in response to systemic inflammation from the interaction between host and pathogen that is characterized by a continuum of clinical manifestations [1]. Acute kidney injury (AKI), a common complication of sepsis, is associated with increased mortality [2], and the pathogenesis of sepsis-associated AKI is different from AKI of other causes [3]. Interestingly, survivors from sepsis-associated AKI carry the risk of developing chronic kidney disease (CKD) and the post-sepsis reinfection is common [4]. Hence, sepsis with pre-existing renal injury is more severe than sepsis in previously normal renal function [5,6,7]. With more favorable outcomes of several chronic diseases due to the supportive care improvement, the incidence of sepsis with impaired renal function is increasing [8].

Erythropoietin (EPO), a 30 kDa glycoprotein from renal peritubular interstitial fibroblasts in response to the cellular hypoxia, binds to erythropoietin receptor (EPOR) and activates JAK2 signaling which increases erythropoiesis through the apoptosis prevention of erythroid progenitor cells. Interestingly, EPO was originally seen as a potential treatment for sepsis-associated anemia because there is low EPO in critically ill patients [9] but high EPO in sepsis [10]. During endotoxemia, EPO reduces AKI through the decreased apoptosis [11] and activation of the β-common receptor [12], leading to an improved survival rate [13]. However, the findings were inconsistent with those of another group [14], possibly due to insufficient doses. Perhaps, a dose-difference among studies and the variety in EPOR that shares the common β-chain with the receptors against IL-3, IL-5 and granulocyte macrophage-colony stimulating factor (GM-CSF) [15] might be responsible for the inconsistency among several studies.

Moreover, the data on EPO effects in sepsis superimposed with pre-existing renal dysfunction are lacking despite the increasing incidence of sepsis in patients with CKD who currently use EPO. Because of the prominent role of the kidney in EPO production, the pre-existing renal injury might affect endogenous EPO production. On the other hand, uremia from pre-existing renal injury might interfere with EPOR and the immunomodulatory functions of EPO along with uremia-induced dysfunctions in other organs [16]. Herein, we tested the effects of EPO on several mouse models of sepsis with pre-existing renal injury, aiming to highlight their immunomodulatory functions and the application aspects, and conducted a retrospective cohort study to support the safety and the efficacy of EPO in accordance with the clinical outcomes in patients with sepsis.

## 2. Materials and Methods

### 2.1. Animal Study and Animal Models

The experimental protocol was approved by the Animal Experimentation Ethics Committee of the Faculty of Medicine, Chulalongkorn University (Bangkok, Thailand, protocol number 018/2561) for animals, following the National Institutes of Health (NIH) criteria and in accordance with the ARRIVE guidelines and regulations for animals. Mice were purchased from Nomura Siam International (Pathumwan, Bangkok, Thailand), and 8-week-old male C57BL/6J mice were originally used in all experiments (Figure 1). Cecal ligation and puncture (CLP) and lipopolysaccharide (LPS; endotoxin) injection [17,18,19] were used to test the impact of EPO and the pre-existing renal injury was induced by bilateral nephrectomy (BiNx) and 5/6 nephrectomy (5/6 Nx) following previous protocols [20].

Briefly, CLP was performed under isoflurane anesthesia through abdominal incision with ligation at 1.5 cm from the cecal tip and punctured twice with a 21-gauge needle. An antibiotic, Primaxin (imipenem/cilastatin; Merck, Kenilworth, NJ, USA), at 10 mg/kg in 100 µL of normal saline (NSS) was subcutaneously administered after CLP and once daily thereafter for 3 days for the survival study. Recombinant human EPO (Recormon^®®^; Roche, Basel, Switzerland) at 1000 or 4000 U/kg was divided into 2 subcutaneous injection doses after CLP and at 6 h after CLP following a previous publication [21]. For the model of sepsis with 5/6Nx (5/6Nx CLP), a representative chronic kidney disease model, the 5/6Nx procedure started with resection of the upper and lower poles of the left kidney followed by right nephrectomy 5 days later through the abdominal incision at 8 weeks prior to the CLP surgery. The adequacy of kidney mass resection was inferred from a ratio of removed fragments of left kidney to right kidney of 0.55–0.72. For the model of sepsis with BiNx (BiNx CLP), BiNx with CLP operation through abdominal incision was performed at 8 weeks after sham surgery (Figure 1). Sham was the abdominal incision only to identify kidneys or cecum. After all of the surgical procedures, mice were monitored every 30 min for 2 h then every 1 h until 6 h post-surgery. In parallel, 10 mg/kg tramadol (Taj pharma, Mumbai, Maharashtra, India) in 100 µL of NSS was subcutaneously administered after the operation, at 6 h post-operation and once daily after that for 3 days post-surgery to attenuate the post-operative pain. The moribund mice with more than 15% weight loss, loss of mobility after a touch stimulation and labored respiration were recognized as the humane endpoints that were immediately sacrificed by cardiac puncture under isoflurane anesthesia. In parallel, the LPS injection model was performed in 16-week-old mice (Figure 1) by intraperitoneal administration of endotoxin (LPS) from *Escherichia coli* 026: B6 (Sigma-Aldrich, St. Louis, MO, USA) at 4 mg/kg. Systolic blood pressure (SBP) was measured by tail cuff plethysmography (IITC Life Scientific Instruments, Woodland Hills, CA, USA) as previously described [22]. The SBP data were an average of 3 measurements with a 10 min interval. Blood was collected through tail vein nicking at specific time points and through cardiac puncture at sacrifice. At sacrifice, several organs were collected and kept at −80 °C before use. For survival observation, the mice were observed twice a day for 96 h, starting at 12 h post-CLP by the trained staff of the animal faculty at the Faculty of Medicine, Chulalongkorn University. Then, mice were weighed daily and the mice with more than 15% loss of weight or in the moribund status, as indicated by labored respiration and non (or less) moving after touch stimulation, were counted as dead and immediately euthanized by cardiac puncture under isoflurane anesthesia. All surviving mice in CLP experiments were sacrificed at 96 h after CLP (or sham) surgery. For the short-term CLP sepsis parameters, all sepsis mice (CLP, 5/6 Nx CLP, BiNx CLP) with or without EPO treatment and sham were sacrificed at 18 h post second surgery (at 16 wks old) (Figure 1). In the LPS injection model, the mice were observed every 15 min for 3 h then every 1 h until 12 h post-LPS. After that, the mice were observed twice daily, starting at 18 h post-injection and the moribund mice (as previously described) were immediately euthanized by cardiac puncture under isoflurane anesthesia. All surviving mice in LPS experiments were sacrificed by cardiac puncture under isoflurane anesthesia at 96 h after injection. For the short-term parameters of LPS sepsis, the mice with LPS or NSS with or without EPO treatment were sacrificed at 6 h post injection (at 16 wks old) (Figure 1). Notably, all mice in both CLP and LPS experiments were sacrificed following these criteria without the mice that were found dead during the observation.

### 2.2. Analysis of Blood and Mouse Organs

Hematocrit and serum creatinine were measured by the microhematocrit method with the Coulter Counter (Hitachi 917; Boehringer Mannheim, Indianapolis, IN, USA) and the QuantiChrom Creatinine Assay Kit (DICT-500; Bioassay, Hayward, CA, USA), respectively. In addition, EPO, serum cytokines (tumor necrosis factor (TNF)-α, IL-6 and IL-10) were measured by ELISA (R & D Systems, Minneapolis, MN, USA). For EPOR, mouse organs were collected to perform quantitative polymerase chain reaction (qPCR) as per a previous publication [18]. Briefly, mouse organs were weighed and prepared for RNA using TRIzol and NanoDrop ND-1000 (Thermo Fisher Scientific, Waltham, MA, USA) and converted into cDNA by a reverse transcription system by SYBR Green master mix (Applied Biosystems, Foster City, CA, USA) based on the ΔΔCT method using *β-actin* as a housekeeping gene (Integrated DNA Technologies, Coralville, IA, USA) (Table 1). For apoptosis detection, in 4 mm thick paraffin embedded mouse organs after 10% formalin fixation, anti-active caspase 3 antibody (Cell Signaling Technology, Beverly, MA, USA) was detected by immunohistochemistry and expressed in positive cells per high-power field (200×).

### 2.3. Experiments on a Macrophage Cell Line

The impact of EPO on the murine macrophage cell line RAW 264.7 was tested. As such, macrophages were cultured in Dulbecco’s modified Eagle’s medium (DMEM) supplemented with 10% heat-inactivated fetal bovine serum (FBS) and penicillin-streptomycin (Thermo Fisher Scientific, Waltham, MA, USA) in 5% carbon dioxide (CO_2_) at 37 °C for 24 h. Then, macrophages at 1 × 10^5^ cells/well were treated with LPS at 10 or 1000 ng/well (33 or 3300 g/L) with or without recombinant EPO (Roche) at 0.5 or 5 µg/well (0.17 or 1.7 g/L) before determination of supernatant cytokines with ELISA assays (R & D Systems) 24 h later. In addition, qPCR was performed to explore *EPOR* expression [18]. Briefly, macrophages at 1 × 10^5^ cells/well were activated by LPS (Sigma-Aldrich) as described above or by recombinant mouse cytokines, including IL-10, IL-6 and TNF-α (R & D Systems), at 5 ng/mL for 6 h. Then, qPCR was performed by SYBR Green master mix (Applied Biosystems, Foster City, CA, USA) based on the ΔΔCT method using *β-actin* as a housekeeping gene (Integrated DNA Technologies, Coralville, IA, USA) (Table 1).

In parallel, Western blot analysis of cells and mouse organs was performed according to a previous publication [18]. In brief, cell lysates in radioimmunoprecipitation assay buffer (RIPA) supplemented with 1× protease inhibitor cocktail (Thermo Fisher Scientific, Waltham, MA, USA) were prepared. Subsequently, samples at 20 µg of total protein, as determined with the bicinchoninic acid assay (Thermo Fisher Scientific, Waltham, MA, USA), were subjected to SDS-PAGE, transferred onto PVDF (polyvinylidene fluoride or polyvinylidene difluoride) membranes, blocked by 5% BSA (bovine serum albumin) in TBS-T (Tris-buffered saline with 0.05% Tween 20) buffer and incubated with specific primary antibodies against mouse EPOR, phosphorylated-NFκB-p65 (p-NFκB), NFκB-p65 (NFκB) or β-actin (Cell-Signaling Technology, Beverly, MA, USA) overnight at 4 °C. After that, the secondary antibody linked with horseradish peroxidase enzyme was used and visualized by ImageQuant™ LAS 500 (GE-Healthcare, Marlborough, MA, USA). Several primary antibodies were added in the same gel, when possible, to have a more reliable analysis; however, the molecular weights of some proteins were too close to perform the analysis in the same gel. Despite some concerns of the cross-reaction between primary antibodies, the Western blot analysis was primarily performed in the same gel.

### 2.4. Experiments on a Hepatocyte Cell Line

To test the direct impact of LPS and inflammatory cytokines on hepatocytes, HepG2 (human hepatoma cells) cells were cultured in modified DMEM for 24 h under the above conditions used for RAW264.7 cells. The cell line was obtained from the American Type Culture Collection (ATCC, Manassas, VA, USA). Then, HepG2 cells at 1 *×* 10^5^ cells/well were treated with LPS (Sigma-Aldrich) or recombinant human cytokines (R & D Systems, Minneapolis, MN, USA) at 5 ng/mL under the above conditions for 24 h before measurement of supernatant cytokines by ELISA kits for human cytokines (R & D Systems, Minneapolis, MN, USA) and Western blot analysis of the cell lysate with antibodies against human EPOR and human *β-actin* (Cell-Signaling Technology) following the above protocol.

### 2.5. General Description of the Human Study

A retrospective cohort study between January 2014 and February 2020 was performed in accordance with STROBE guidelines and regulations for humans. Databases and records between November 2020 to May 2021 were collected. The experimental protocol was approved by the Ethics Committee of the Faculty of Tropical Medicine, Mahidol University (MUTM 2020-040-01), and registered in the Thai Clinical Trials Registry (TCTR20201022004). Written informed consent was obtained from participants. All eligible participants were diagnosed as sepsis or septic shock and treated in the medical intensive care units (ICUs) following the sepsis bundle guideline [23]. Sepsis was defined as life-threatening organ dysfunction caused by a dysregulated host response to infection [1]. Septic shock was defined by the persistence of hypotension requiring vasopressors to maintain a mean arterial pressure (MAP) of ≥65 mmHg and a serum lactate level >2 mmol/L (18 mg/dL) despite adequate volume resuscitation [1]. Other inclusion criteria were patients at 18 years or older who received EPO due to their underlying diseases. The exclusion criteria were the incomplete data, participants referred to other hospitals, participants diagnosed within 3 months prior to the recruitment with ischemic heart disease, ischemic stroke, embolism, thrombosis, hypercoagulable disorder and gastrointestinal bleeding. The vasopressor dependency index (VDI) was used to determine the relationship between the dose of vasopressor (units/min) and MAP as follows: VDI = ((dobutamine dose *×* 1) + (dopamine dose *×* 1) + (norepinephrine dose *×* 100) + (vasopressin dose *×* 100) + (epinephrine dose *×* 100))/MAP [24], and were expressed as μg/kg/min. In the present study, low-dose, high-dose and very-high-dose EPO is defined as EPO doses < 8000, 8000–16,000 and >16,000 units/week, respectively, following a continuous erythropoietin receptor activator (CERA) publication [25]. For the study outcome, primary endpoint was the 29-day mortality rate and secondary endpoints were vasopressor exposure (VDI), percentage of the patients with red-cell transfusion, the alteration in hemoglobin (Hb) concentrations and EPO adverse effects.

### 2.6. Statistical Analysis

Continuous variables are expressed as the mean ± standard error (SE) in the mouse study and, in the human study, quantitative data are summarized as the mean ± SE or standard deviation (SD) for normally distributed variables, or median and interquartile ranges (IQRs) for nonnormally distributed variables, and qualitative data are presented as n (percentage). To compare means, in the animal study, Student’s *t*-test or one-way analysis of variance (ANOVA) with Tukey’s comparison test was used for the analysis of experiments with 2 and more than 2 groups, respectively, and the survival analysis was performed by the log-rank test. In the patients, the mortality in the primary endpoint was analyzed with the Kaplan–Meier method. The Cox proportional hazards regression model was applied to calculate the hazard ratio (HR) and associated 95% confidence interval (95% CI). The final analyses were based on a two-sided significance level of 0.05. The *p* value < 0.05 was considered statistically significant. The statistical analyses were performed by IBM SPSS version 23.0 (IBM Corp, Armonk, NY, USA), Microsoft Excel (2016, Microsoft Corporation, Redmond, WA, USA) and GraphPad Prism 8.4.3 (GraphPad Software, La Jolla, CA, USA).

## 3. Results

Sepsis, in all tested mouse models, enhanced the endogenous serum EPO. The necessity of a high dose of EPO for its anti-inflammatory effect in sepsis might be due to the reduced EPOR in macrophages.

### 3.1. Sepsis Enhanced Endogenous EPO Production with or without Renal Injury Model, and Reduced EPO Receptor in the Liver

Sepsis is a potent inducer of EPO from kidneys, as demonstrated by increased serum EPO in both the cecal ligation and puncture (CLP) and LPS models (Figure 2A upper). The baseline value of serum EPO at 8 weeks after 5/6Nx, a model of chronic kidney disease, without sepsis was higher than non-Nx mice (Figure 2A middle and lower) despite the lower renal mass in 5/6Nx mice. Serum EPO in 5/6Nx + CLP mice was higher than 5/6Nx + sham but was still lower than serum EPO in mice with normal kidney CLP (Figure 2A upper and middle). The increased serum EPO after LPS injection was lower than non-Nx CLP mice (Figure 2A upper). Surprisingly, serum EPO was mildly elevated at 18 h after CLP surgery in BiNx mice (Figure 2A lower), highlighting an impact of hepatocyte EPO production. However, *EPOR* expression and the protein abundance of EPOR in sham control mice was detectable mostly in the liver and decreased at 18 and 6 h after CLP surgery and LPS injection, respectively (Figure 2B,C) which, at least in part, was due to hepatocyte NFκB-mediated apoptosis (Figure 2D–F). Indeed, CLP induced apoptosis in the kidney and spleen, but not in the heart, while LPS caused apoptosis only in the spleen (Figure 2E,F). Hence, the increased EPO production in sepsis is, partly, a compensation for the reduced EPOR in sepsis and implies a possible influence of EPO on sepsis.

### 3.2. Mortality, Pre-Conditioning Renal Injury and Inflammation in Sepsis Were Attenuated by High-Dose EPO

Indeed, high-dose EPO, but not low-dose, improved sepsis survival in all models, except for LPS, regardless of the preconditioning kidney injury (Figure 3A–D) without an effect on SBP (Figure 3E–H). Although EPO did not increase Hct in CLP mice (non-Nx and 5/6Nx), Hct in CLP mice with high-dose EPO was not different from sham control mice, implying a partial erythropoiesis effect of EPO (Figure 3I,L). Notably, there was no effect of EPO on Hct in BiNx and LPS models (Figure 3K,L). Furthermore, high-dose EPO, but not low-dose, attenuated sepsis-induced renal injury in CLP with or without 5/6Nx (Figure 4A–H) but not in the LPS model, possibly due to the non-renal injury in the LPS model (Figure 4D,H). However, EPO did not attenuate blood bacterial count (Figure 4I–K).

Interestingly, while high-dose EPO attenuated serum cytokines in all sepsis models (Figure 5), low-dose EPO reduced only serum TNF-α and IL-10 in BiNx-CLP mice (Figure 5C and Figure 2A). Among CLP mice, the mice without pre-existing renal injury (No Nx) demonstrated the least severity, while BiNx CLP mice showed the highest severity as determined by serum cytokines (Figure 5A–K). However, spleen apoptosis of EPO-treated CLP in No Nx mice and BiNx mice was evaluated and EPO attenuated spleen apoptosis in both models, regardless of the different in the severity of sepsis (Figure 6A,B).

### 3.3. EPO Attenuated Inflammatory Responses in LPS-Activated Macrophages

The EPO anti-inflammatory property was demonstrated in vitro. At low LPS concentrations (10 ng/well), EPO at both high and low concentrations, 5 and 0.5 µg/well (0.17 or 1.7 g/L), respectively, attenuated supernatant TNF-α and IL-6 but not IL-10 levels (Figure 7A–C). On the other hand, with high-dose LPS stimulation, 1000 ng/well (3300 g/L), only high concentrations of EPO reduced supernatant cytokines (Figure 7D–F). Interestingly, high-dose EPO reduced the expression of *iNOS* and *IL-1β*, the markers of proinflammatory M1 macrophage polarization, after LPS stimulation (Figure 8A,B). Meanwhile, for anti-inflammatory M2 macrophage polarization, LPS alone reduced *Fizz-1*, but not *Arginase-1*, and EPO incubation with LPS enhanced only *Arginase-1* expression but not *Fizz-1* (Figure 8C,D). In parallel, LPS increased the expression of *TLR-4*, *STAT3* and *NF-κB* and the abundance of phosphorylated-NF-κB, the LPS receptor and downstream signaling, respectively, which were reduced by a high EPO concentration (Figure 8E–H).

Furthermore, the direct impact of LPS and inflammatory cytokines on EPOR was tested in cell lines because of the low EPOR abundance in all organs of CLP sepsis mice (Figure 2D,E). Accordingly, incubation with LPS, but not inflammatory cytokines, in macrophages reduced *EPOR* expression (Figure 9A,B), and LPS also reduced EPOR abundance in macrophages (Figure 9C). In contrast, the impact of LPS and inflammatory cytokines on hepatocytes was different, as both LPS and inflammatory cytokines enhanced *EPOR* expression and EPOR abundance in the hepatocyte cell line (Figure 10A,B), without an alteration in hepatocyte EPO production (Figure 10C,D).

### 3.4. Mortality Rate Was Lower in Patients with EPO Administration, but Thrombotic Vascular Adverse Events Was Not Significantly Increased

Because of the possible anti-inflammatory effect of EPO from the mouse models, a retrospective study in patients was performed to explore the possible correlation between EPO administration and sepsis outcomes. Accordingly, a total of 344 participants with sepsis were included in this study, of whom 151 patients received EPO (EPO group) and 193 patients did not receive EPO (Figure 11) as demonstrated by data in Table 2 and Table 3. However, patients in the EPO group were older than those in the non-EPO group (65 ± 13 vs. 57 ± 8 years) (mean ± SD) with more underlying diseases, including chronic kidney disease (CKD). The major cause of anemia in the EPO group was CKD and iron deficiency anemia. Additionally, 6.2% in the EPO group received only one dose of EPO during the study period. While low-dose EPO (<8000 units/week) was prescribed in 32% of patients, high-dose EPO (8000–16,000 units/week) was prescribed in 67% of patients treated with EPO (Table 2).

Despite EPO administration in the EPO group, there was a significantly higher number of patients with red-cell transfusion compared with the non-EPO group (42% vs. 26%; odds ratio, 2.1; 95% confidence interval (CI), 1.33 to 3.32; *p* = 0.002) without the difference in total units of blood and transfusion complications (Table 4). Regarding vasopressor exposure, vasopressor dependence index (VDI; see Section 2) decreased as early as 3 days of the sepsis (Table 4), which potentially contributed to a higher proportion of patients recovering from sepsis in the first 5 days of treatment (11% in the EPO group (from n = 45 as of day 1 to n = 40 as of day 5) vs. 6.3% in the non-EPO group (from n = 64 as of day 1 to n = 60 as of day 5) (Table 4).

As shown in Table 4, the cause of death from multiple organ failure was significantly lower in the EPO group than the non-EPO group. Interestingly, the 29-day mortality was significantly lower in the EPO group than the non-EPO group (15% in the EPO group vs. 23% in the non-EPO group, *p* = 0.04), according to the Kaplan–Meier estimation (Figure 12A). Moreover, in the subgroup analysis, mortality at day 29 was significantly lower in the high-dose EPO group when compared with the low-dose EPO (7% vs. 31%, *p* < 0.0001) and the non-EPO group (7% vs. 23%, *p* < 0.001). Indeed, the higher the EPO doses, the lower the VDI was demonstrated (Figure 12B,C). There was a non-significant increase in the incidence of thrombotic vascular events between the EPO group vs. the non-EPO group and between EPO doses > 8000 vs. < 8000 units/week (Table 2 and Table 3).

Cox proportional hazards regression analysis showed increased mortality for each of the following characteristics in patients hospitalized for sepsis receiving EPO compared with those without EPO (Table 5): male gender (HR 1.13, 95% CI 1.02–1.18), older age (age 65–69 vs. <65 years: HR 1.79, 95% CI 1.45–1.89; age 70–79 vs. <65 years: HR 2.17, 95% CI 2.02–2.28), diabetes mellitus (HR 1.17, 95% CI 1.10–1.20) and renal function determined by estimated glomerular filtration rate (eGFR) (eGFR 15–29 vs. >60 mL/min/1.73 m^2^ (HR 1.62, 95% CI 1.34–1.70); eGFR < 15 vs. >60 mL/min/1.73 m^2^ (HR 1.18, 95% CI 1.14–1.22)). Nevertheless, the EPO group showed lower HR compared to the non-EPO group, particularly in those with eGFR 15–29 mL/min/1.73 m^2^. This may be as a result of EPO prescription in the EPO group.

## 4. Discussion

A non-erythropoiesis anti-inflammatory effect of EPO was demonstrated through (i) the attenuation of sepsis with pre-existing renal injury mouse models similar to the effect on sepsis without pre-existing injury [12] and (ii) the retrospective data in patients supporting EPO as a sepsis adjunctive therapy.

### 4.1. The Increased EPO Production in Response to the Reduced EPOR during Sepsis, an Immunomodulation of EPO

There was non-significant different between normal mice and the mice with kidney injury at 0 time point despite a tendency of a lower EPO level in normal mice, implying an influence of systemic inflammation on an increase in EPO production [27]. Similarly, increased EPO production during sepsis has been previously mentioned, possibly due to decreased systolic blood pressure (SBP)-induced renal hypoxia [28]. Additionally, CLP surgery, but not LPS injection, also induced acute anemia, similar to the previous publications [29,30], possibly due to the dilutional effect of post-operative fluid administration or sepsis-induced occult bleeding in the internal organs [31]. Nevertheless, EPO significantly restored the hematocrit in 5/6Nx CLP mice and attenuated sepsis severity, possibly through the improved tissue hypoxia. Although most of the mice died at 18 h after 5/6 Nx CLP, the EPO levels of some surviving mice at 24 h post-CLP were not different from 18 h post-CLP (data not shown). Moreover, sepsis induced EPO production in mice with a limited renal mass (5/6Nx), despite being at a lower level than sepsis in normal kidneys and possibly induced hepatic EPO as increased EPO in mice with removed kidneys (BiNx). Because of the pleiotropic effects of EPO [32,33,34], increased EPO production during sepsis is not only due to renal hypoxia but possibly also a response toward EPOR reduction. While LPS administration (but not incubation with cytokines) prominently reduced EPOR in macrophages, LPS and cytokines mildly increased EPOR in hepatocytes (Figure 8C and Figure 9A,B). Despite the discrepancy between increased hepatocyte-EPOR and reduced macrophage-EPOR, the total expression of *EPOR* in livers of sepsis mice, a combination of hepatocytes and macrophages (Kupffer’s cells) in liver, was lower than the control group, supporting a possibly too-low EPOR during sepsis. Although LPS and inflammatory cytokines enhanced EPOR in hepatocytes in our in vitro data, the decreased viable hepatocytes due to sepsis-induced hepatocyte apoptosis [35] and the increased inflammatory macrophages in livers during sepsis might be responsible to the lower total *EPOR* in sepsis livers. Despite the detection of EPOR in several immune cells, either innate immunity (macrophages, dendritic cells and mast cells) or adaptive immunity (T cells and B cells) [36], macrophages are important for sepsis immune response, especially in LPS and CLP models [18,37]. Because EPO protects several organs, including the liver, through activation of the EPO receptor/β-common receptor (EPOR/βcR) complex [32,33,34], the inadequate expression of *EPOR* during sepsis possibly reduced the protective effects of EPO. Additionally, EPO also protects several cells from excessive autophagy in several models [38,39]. Hence, EPO enhanced erythropoiesis through the anti-apoptosis of erythroid progenitor cells [33] and attenuated apoptosis in the liver and kidney [12]. Because of several non-erythropoiesis properties of EPO [40], increased endogenous EPO production in sepsis might not only be a response to renal hypoxia but also for the other protective purposes.

Likewise, several non-erythropoiesis effects of EPO, especially anti-apoptosis, are well known, as (i) EPOR knockdown [33] or soluble anti-EPOR induces cell apoptosis [41], (ii) EPO attenuates lymphoid apoptosis in several organs, including the spleen, lymph node and intestine, in a CLP rat model [42] and (iii) EPO improves sepsis-induced multiorgan injuries [43,44]. Here, EPOR activation downregulated *NF-ΚB* through reduced *TLR-4* and *STAT3**,* which inhibited cytokine storm in sepsis, as illustrated in Figure 13, supporting previous publications [45,46,47,48,49]. In parallel, EPOR activation downregulated *iNOS* and *IL-1β* but increased *Arginase-1* and shifted macrophage polarization into an anti-inflammatory stage, to properly balance against sepsis proinflammation [50,51]. Here, gene expression and protein abundance of EPOR in liver and macrophages was reduced by inflammatory sepsis (LPS and CLP) and LPS, respectively, while LPS increased EPOR expression in hepatocyte cell lines. Hence, there was a different role of EPO between immune cells and hepatocytes. In hepatocytes, both LPS and inflammatory cytokines increased EPOR, perhaps trying to increase hepatic EPO production during sepsis for counteracting against hypotension and/or hepatocyte apoptosis. As such, increased EPO production and erythrocytosis are demonstrated in patients with hepatocellular carcinoma [52] and relatively increased EPOR expression with decreased EPO level by LPS treatment is possible. In contrast, LPS reduces *EPOR* in macrophages, possibly due to a limited ability of macrophages to express *EPOR*. Hence, a supraphysiologic dose of EPO is necessary for macrophage immunomodulation and is beneficial for both macrophages and hepatocytes. Despite a non-difference on *EPOR* expression in kidneys between sepsis and control mice here, an impact of EPOR is beneficial for renal tubular cell protective effect [53], which might be different from macrophages and hepatocytes. More studies on this topic would be interesting.

### 4.2. High-Dose EPO Administration in Mice and in Patients, an Adjuvant Therapy for Sepsis

An increased dose of EPO is necessary to maintain hematocrit (Hct) during septicemia [54], perhaps due to sepsis-reduced EPOR. During CLP sepsis, serum EPO level increased up to 20–40-fold and 5–8-fold in mice with normal kidney and low renal mass (5/6Nx), respectively, which still was not enough to maintain Hct in CLP mice. These data imply the increased EPO requirement and a limitation of EPO production during sepsis. This also supports the requirement of the high-dose EPO for sepsis immunomodulation [55]. Because of polycythemia, a possible side effect of high-dose EPO in sepsis, a low dose of EPO was also tested here. Unfortunately, only the high dose of EPO (4000 units/kg) attenuated CLP sepsis regardless of the preconditioning renal injury. Interestingly, EPO attenuated sepsis in CLP mice with kidney removal, supporting the extrarenal beneficial effect of EPO during sepsis. However, the benefit of EPO in sepsis in our models did not depend on erythropoiesis or improved cardiovascular effects as previously mentioned [12,43], perhaps because the EPO administration was too short to show these effects. Moreover, the high-dose EPO also attenuated serum cytokines in the LPS model without renal injury, implying a direct anti-inflammatory effect of EPO against LPS. Here, the short course of EPO did not cause polycythemia, a possible EPO adverse effect in sepsis, in all of these models. Hence, high-dose and short-course EPO administration in sepsis is an interesting sepsis adjuvant therapy.

In patients, EPO showed beneficial effects on the 29-day survival rate, different from the two previous randomized trials, probably due to the distinctly larger CKD population in our study; stage 3–5 CKD here vs. stage 1–2 CKD in the previous publications [56,57]. Despite the EPO resistance from uremia, EPO effectively maintained Hb concentration and attenuated sepsis with the comparable red-cell transfusions even with the inferior renal function compared with the non-EPO group (Table 2 and Table 4). Indeed, high-dose EPO (>8000 units/week), but not the low-dose EPO, showed the better sepsis outcomes including survival rate and amount of vasopressor—the greater the EPO dose, the lesser the vasopressor dependence index (VDI) (Figure 12) due to the sepsis-induced EPO resistance [54,58]. Although the majority of patients in the EPO group received a more frequent blood transfusion, there were no significant difference in units of the transfused blood per patient between EPO and non-EPO groups. Additionally, there was non-alteration in Hb concentration between the baseline and at day 29 in the EPO group, despite a more aggressive blood transfusion than the non-EPO group, implying an inadequate blood transfusion in the patients of the EPO group. Moreover, the patients in the EPO group were older with lower Hb concentration at the baseline when compared with the non-EPO group. Despite these in-advantage factors in the EPO group (older age with more co-existing conditions and anemia with inadequate blood transfusion), there was a better survival rate in the EPO group. Therefore, EPO might be beneficial in survival improvement as other interventions for sepsis treatment were comparable between EPO and non-EPO groups. Furthermore, the baseline characteristics of the patients with high-dose EPO were comparable to the group with low-dose EPO (Table 3), supporting the benefit of EPO in the high-dose EPO group. Notably, the mortality rate of the patients with the low-dose EPO did not differ from the non-EPO group, despite a tendency of a higher mortality in the low-EPO group, possibly due to the more complicate underlying diseases in this group. In contrast to the previous studies [56,57], EPO administration before sepsis might be beneficial in the early phase of sepsis or septic shock, as demonstrated by the lower VDI, at least during the first 72 h of sepsis (Table 4). Perhaps, high-dose EPO might not only intervene the process of multiorgan dysfunction due to sepsis (Table 4) but also ameliorate the course in less reserved renal function backgrounds (Table 5), consistent with the observation that hemodialysis patients with COVID-19—a sepsis-like syndrome—are likely to develop less severe pneumonia [59]. Despite the concerns on EPO-induced thrombotic events, thrombotic vascular events were not increased in the present study, perhaps due to the non-polycythemia in our patients. However, several limitations should be mentioned. First, the retrospective cohort did not include several anemia biomarkers, including circulating EPO levels, vitamin B12, serum folate and haptoglobin. Second, there was a lack of a relationship between EPO levels in accordance with EPO dose and sepsis status. Third, the differences in baseline characteristics, the number of patients between groups and the limited sample size were also limitations. Further studies on high-dose EPO in sepsis are needed.

## 5. Conclusions

Our findings demonstrated the sepsis attenuation by the high-dose EPO in either experimental sepsis or patients through EPO anti-inflammatory properties, as illustrated in Figure 13. The necessity of high-dose EPO is partly due to the LPS-induced EPOR downregulation in sepsis. Because of the availability of EPO in current clinical practice, we proposed a short-course and high-dose EPO as an adjuvant therapy for sepsis. Further prospective controlled studies are warranted.

## Figures and Tables

**Figure 1 cells-10-03133-f001:**
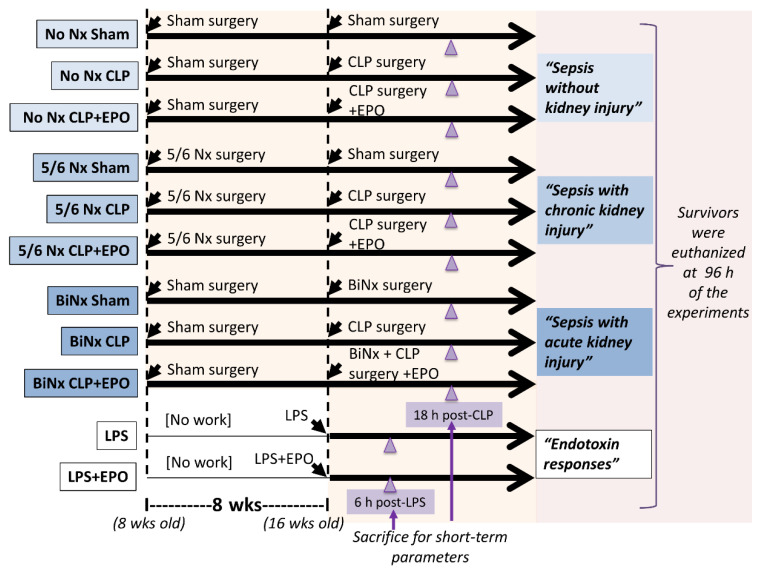
A schematic of the experimental groups is shown. For the short-term parameters of sepsis in LPS and CLP models, mice were sacrificed at 6 and 18 h of the experiments, respectively. Otherwise, the surviving mice were euthanized at 96 h of the experiments following the animal study protocol. BiNx, bilateral nephrectomy; CLP, cecal ligation and puncture; EPO, erythropoietin administration; LPS, lipopolysaccharide.

**Figure 2 cells-10-03133-f002:**
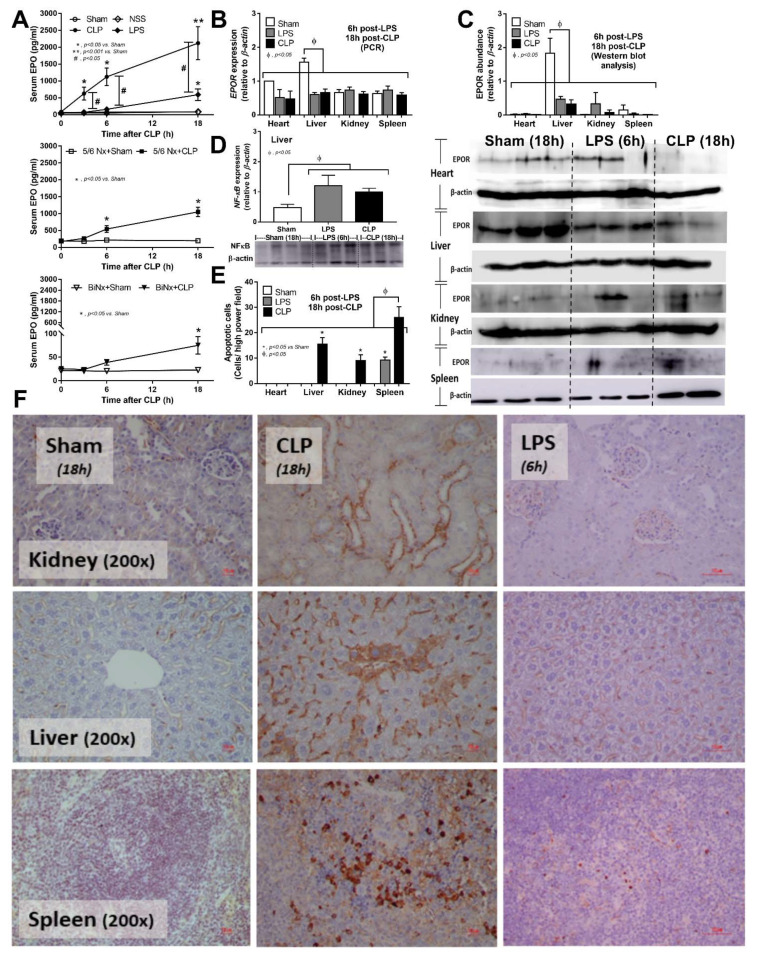
Effects of sepsis models on erythropoietin (EPO) and EPO receptor (EPOR). Time course of serum EPO in mice without pre-existing renal injury after cecal ligation and puncture (CLP) or control surgery (sham) and LPS injection (LPS) or control normal saline (NSS) injection (**A upper**), in mice with 5/6 nephrectomy (5/6Nx) and CLP (5/6Nx + CLP) or sham surgery (5/6Nx + sham) (**A middle**) and in mice with bilateral nephrectomy (BiNx) and CLP (BiNx + CLP) or sham (BiNx + sham) (**A lower**) are demonstrated (n = 8–10/time point). Additionally, gene expression of EPO receptor *(EPOR)* by polymerase chain reaction (PCR) **(B)**, EPOR protein abundance with the representative Western blot analysis in several organs (**C**), NFκB abundance in liver (**D**) and apoptotic cells with the representative pictures in several organs (as evaluated by anti-activated caspase 3 immunohistochemistry staining; original magnification 200×) (**E**,**F**) from mice after 18 h post-CLP (or sham) or at 6 h post-LPS are demonstrated (n = 4–7/group for (**B**,**E**)). The representative pictures of mice with NSS injection are not demonstrated due to the similarity to sham surgery control mice.

**Figure 3 cells-10-03133-f003:**
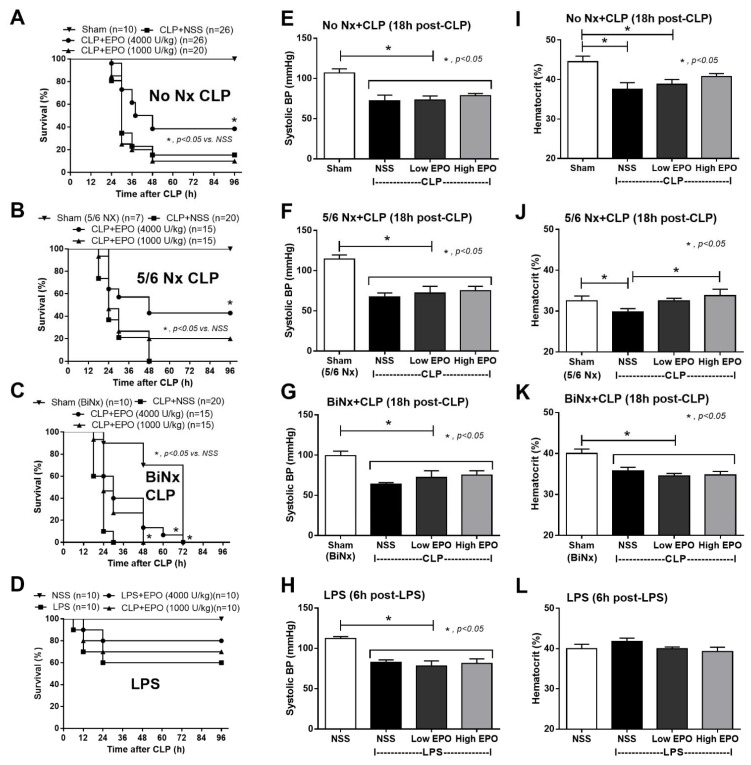
Erythropoietin (EPO) administration attenuates sepsis severity in sepsis mouse models. Characteristics of cecal ligation and puncture (CLP) mice without renal resection (no Nx CLP) or with 5/6 nephrectomy (5/6Nx) or bilateral nephrectomy (BiNx) and lipopolysaccharide (LPS) administered to mice after treatment with EPO or normal saline (NSS) as determined by survival analysis (**A**–**D**), systolic blood pressure (**E**–**H**) and hematocrit (**I**–**L**) are demonstrated (n = 7–10/group for (**E**–**L**)).

**Figure 4 cells-10-03133-f004:**
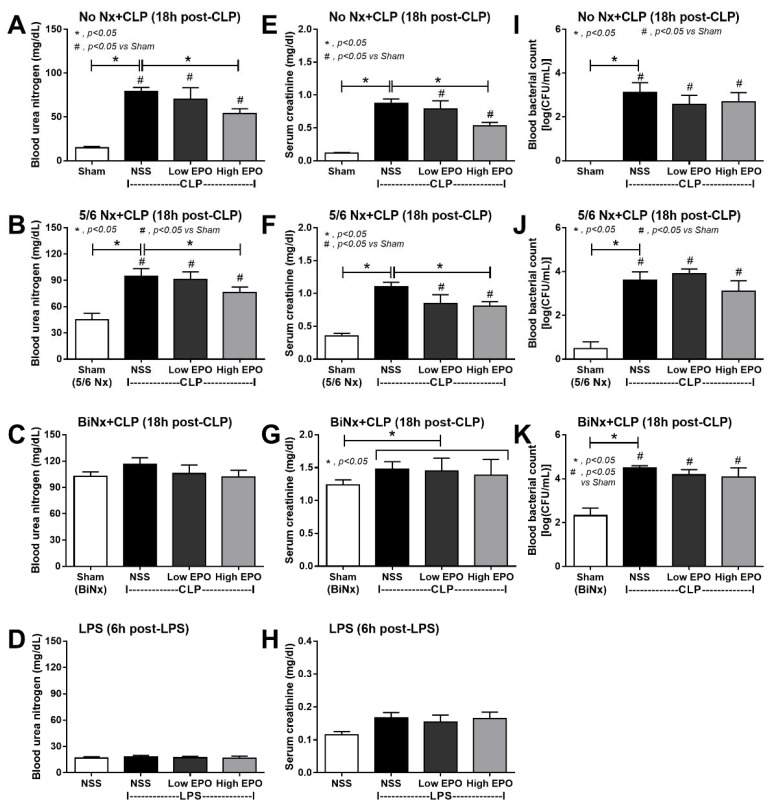
Erythropoietin (EPO) administration attenuates sepsis-induced renal injury and bacteremia in sepsis mouse models. Characteristics of cecal ligation and puncture (CLP) mice without renal resection (no Nx CLP) or with 5/6 nephrectomy (5/6Nx) or bilateral nephrectomy (BiNx) and lipopolysaccharide (LPS) administered to mice after treatment with EPO or normal saline (NSS) as determined by blood urea nitrogen (BUN) (**A**–**D**), serum creatinine (**E**–**H**) and blood bacterial count (**I**–**K**) are demonstrated (n = 7–10/group for (**A**–**K**). Notably, bacteremia was not detectable in mice at 6 h post-LPS (data not shown).

**Figure 5 cells-10-03133-f005:**
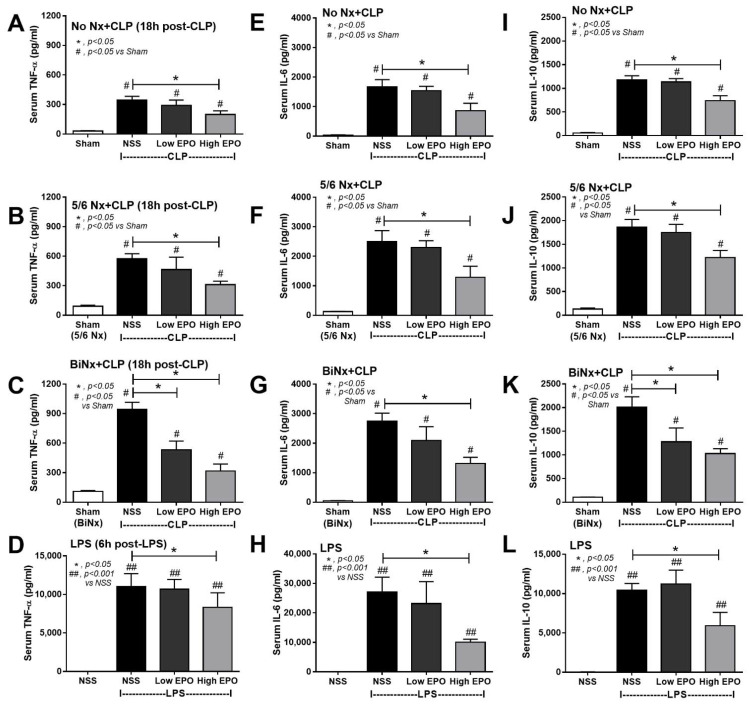
Erythropoietin (EPO) administration attenuates serum cytokines in sepsis mouse models. Serum TNF-α (**A**–**D**), IL-6 (**E**–**H**) and IL-10 (**I**–**L**) in cecal ligation and puncture (CLP) mice without renal resection (no Nx CLP) or with 5/6 nephrectomy (5/6Nx) or bilateral nephrectomy (BiNx) and lipopolysaccharide (LPS) administered to mice after treatment with EPO or normal saline (NSS) (**A**–**D**) are demonstrated (n = 7–10/group).

**Figure 6 cells-10-03133-f006:**
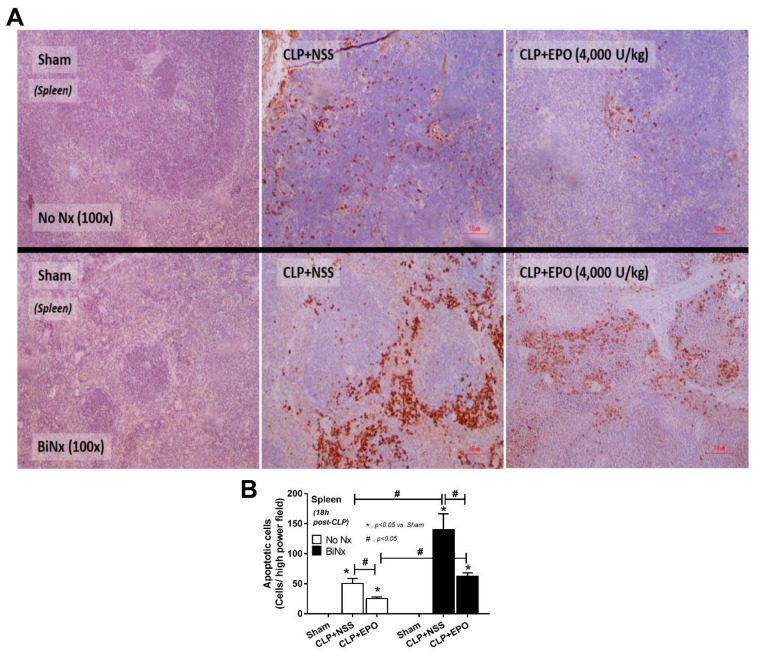
Erythropoietin (EPO) administration attenuates spleen apoptosis in sepsis mouse models. Apoptotic cells in spleen of mice after cecal ligation and puncture (CLP) after treatment with EPO or normal saline (NSS) versus sham mice with the representative pictures of activated caspase 3 immunohistochemistry staining (**A**) and quantification analysis (**B**) are demonstrated (n = 3–6/group).

**Figure 7 cells-10-03133-f007:**
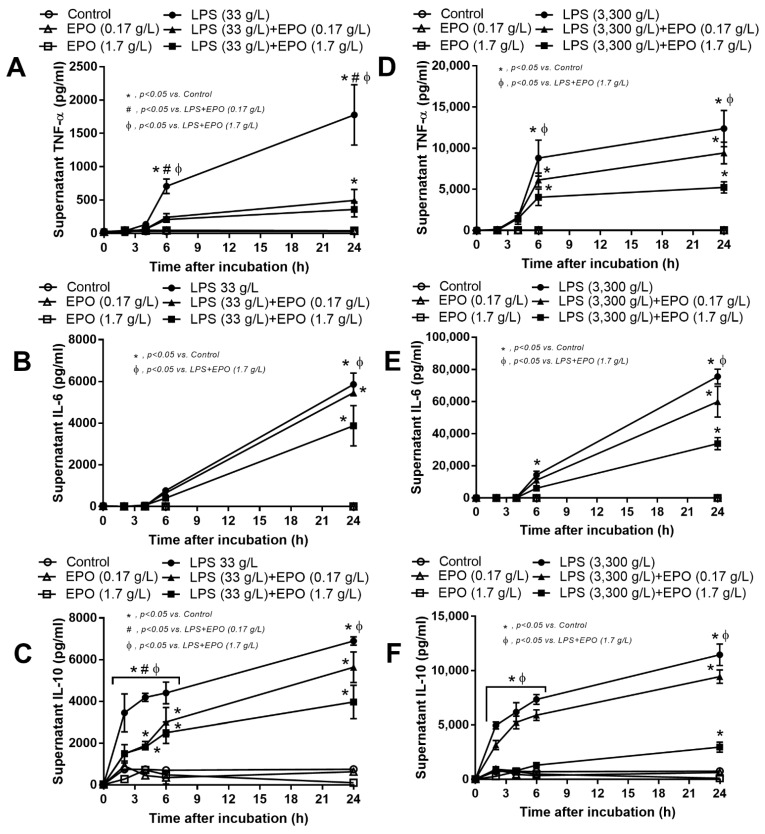
Erythropoietin (EPO) attenuates supernatant cytokines in lipopolysaccharide (LPS)-activated macrophages. Time courses of supernatant cytokines from macrophages with 6 h of control culture media (control), EPO alone (EPO), LPS alone (LPS) or EPO with LPS (LPS + EPO), at low concentration (33 g/L) (**A**–**C**) and high concentration (3300 g/L) (**D**–**F**) are demonstrated (independent triplicate experiments were performed).

**Figure 8 cells-10-03133-f008:**
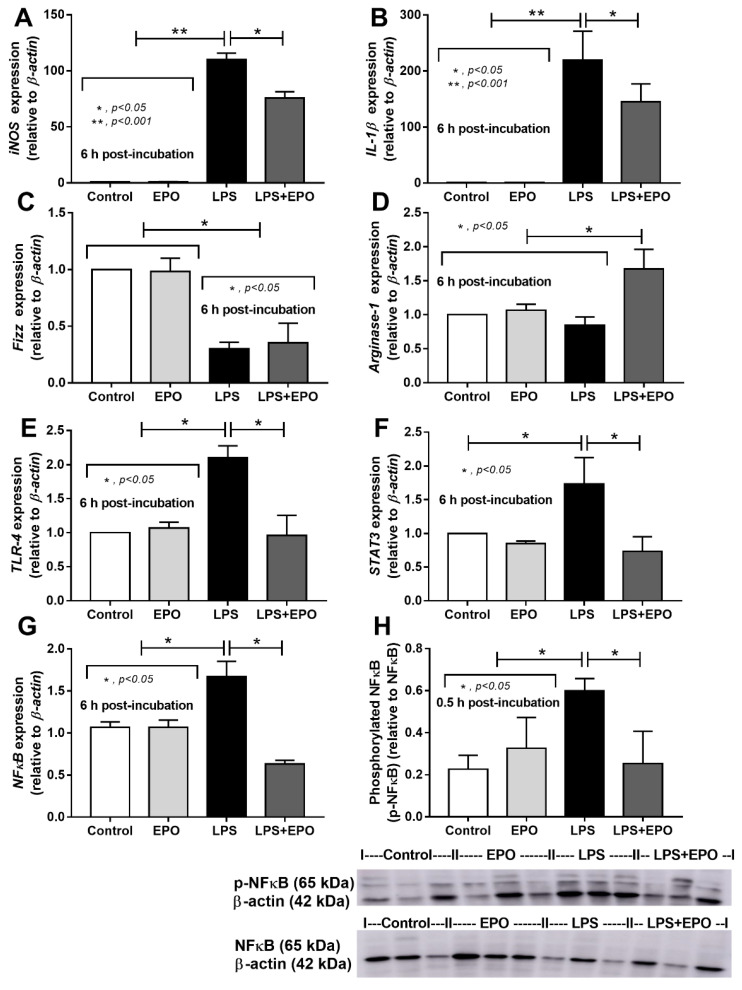
Erythropoietin (EPO) attenuates the expression of several proinflammatory genes in lipopolysaccharide (LPS)-activated macrophages. Expression of genes of (i) proinflammatory macrophage markers: inducible nitric oxide synthase (*iNOS*) and interleukin-β (*IL-1β*) (**A**,**B**), (ii) anti-inflammatory macrophage markers: resistin-like molecule alpha (*Fizz-1*) and *arginase-1*, (**C**,**D**) and (iii) LPS proinflammatory signaling: toll-like receptor-4 (*TLR-4*) (**E**), signal transducer and activator of transcription 3 *(STAT3)* (**F**) and nuclear factor-kappa B (*NFκB*) with the abundance of phosphorylated-NFκB (p-NFκB) by Western blot analysis (**G**,**H**) from macrophages with 6 h of control culture media (control), EPO alone (1.7 g/L) (EPO), LPS alone (3300 g/L) (LPS) or EPO with LPS (LPS + EPO) is demonstrated (independent triplicate experiments were performed).

**Figure 9 cells-10-03133-f009:**
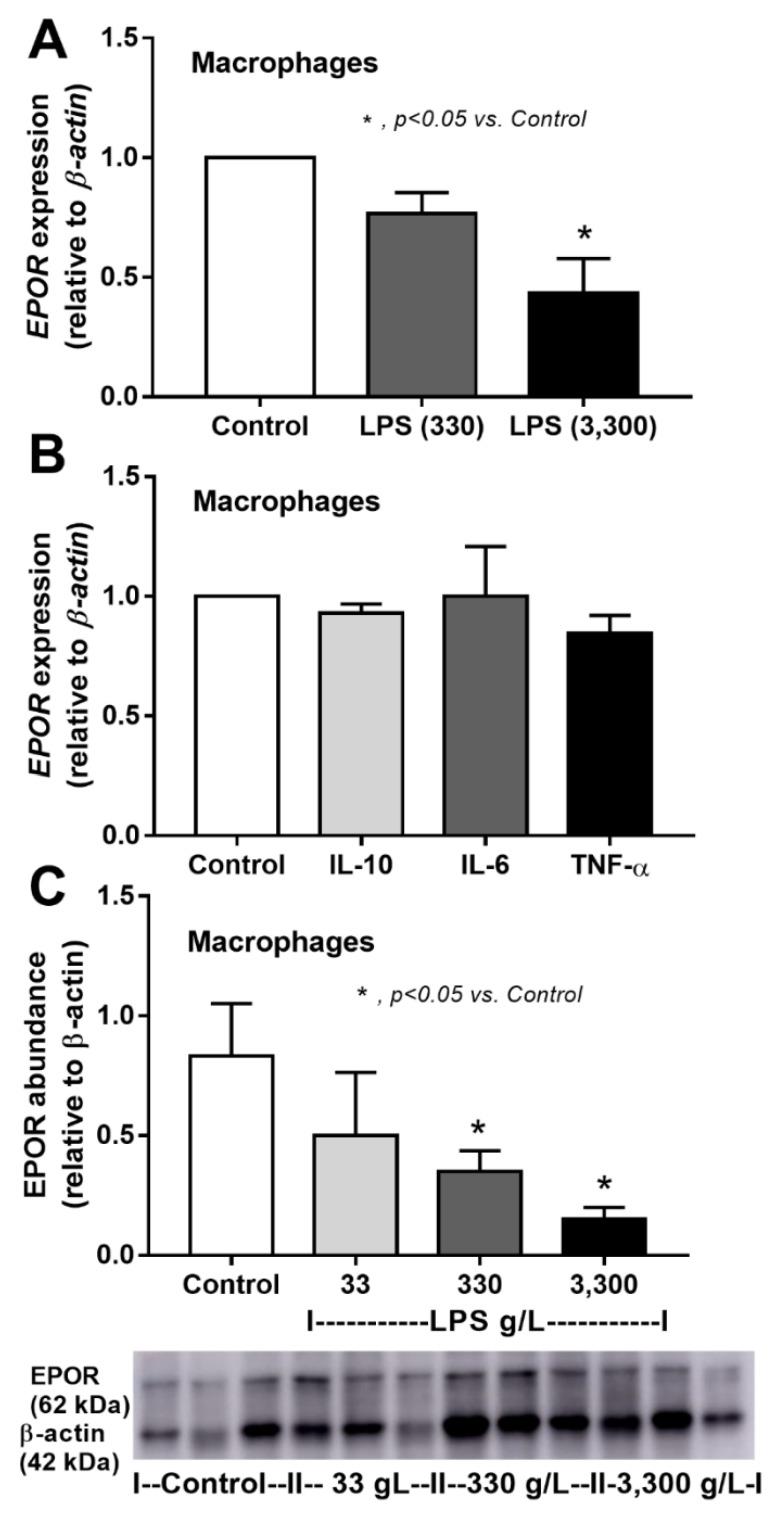
Lipopolysaccharide (LPS) downregulates the expression of the erythropoietin receptor (*EPOR*) gene and reduces EPOR abundance in macrophages. Expression of erythropoietin receptor (EPOR) in macrophages 6 h after culture media (control), LPS at 330 g/L or LPS at 3300 g/L (**A**) or 6 h after activation of different cytokines (**B**) is demonstrated (independent triplicate experiments were performed). In addition, the protein abundance of EPOR in macrophages with 6 h of stimulation by control media or LPS is shown in arbitrary units and representative Western blot analysis (**C**) (independent triplicate experiments were performed).

**Figure 10 cells-10-03133-f010:**
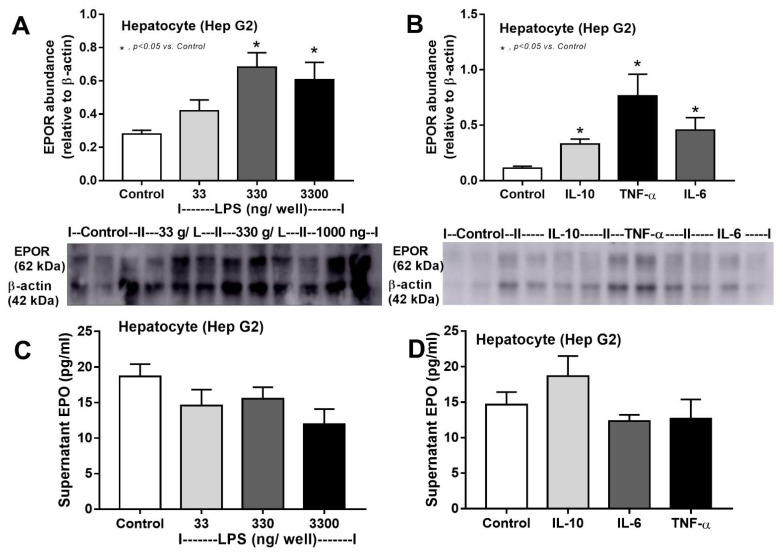
Lipopolysaccharide (LPS) and inflammatory cytokines increase the abundance of erythropoietin receptor (EPOR) in hepatocytes. Protein abundances of EPOR in hepatocytes (HepG2 cell line) 24 h after treatment with culture media (control), LPS at 33–3300 g/L (**A**) or 24 h after activation of different cytokines (**B**) are demonstrated with representative Western blot analysis (independent triplicate experiments were performed). In addition, supernatant erythropoietin (EPO) from hepatocytes after these stimulations (**C**,**D**) is shown (independent triplicate experiments were performed).

**Figure 11 cells-10-03133-f011:**
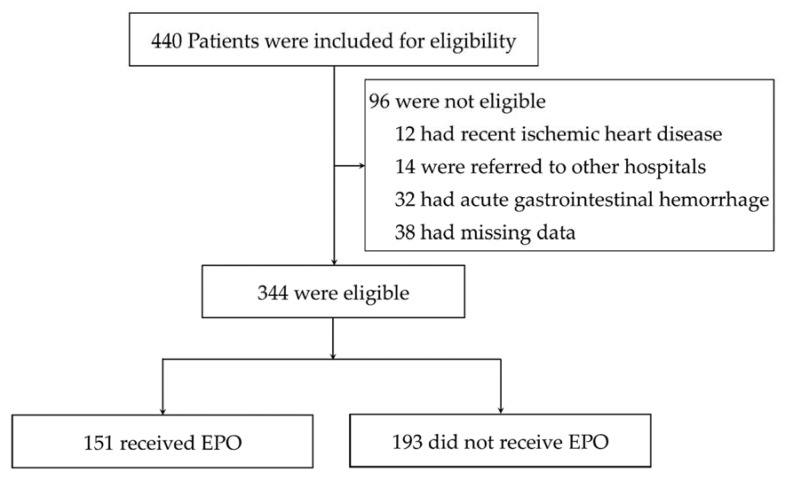
Screening and enrollment of study in patients. EPO denotes erythropoietin.

**Figure 12 cells-10-03133-f012:**
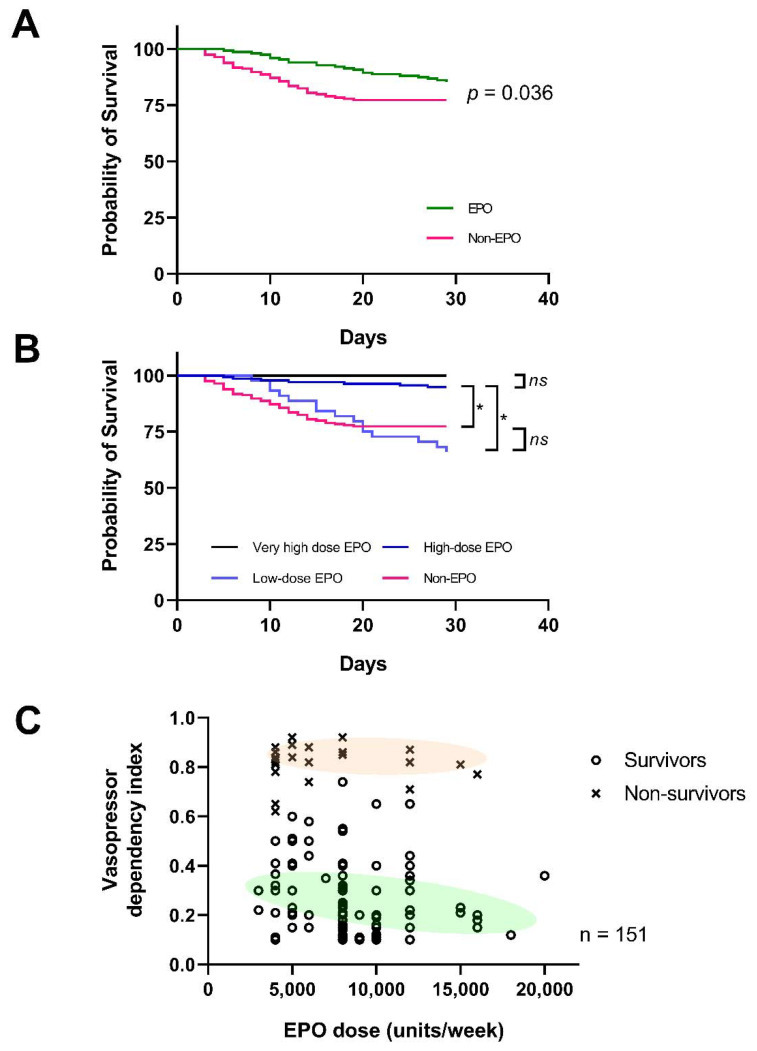
Clinical outcomes of the study. Relationship between erythropoietin (EPO) dose and the vasopressor dependency index (VDI) of patients who received EPO stratified by survivors (n = 129) vs. non-survivors (n = 22), where the green and red shades represent the trends of each group, respectively, formulated by nonlinear fit (**A**) and Kaplan–Meier analysis of mortality through day 29 (**B**) are demonstrated. Data were censored for 129 patients in the EPO group (85.4%) and 149 in the non-EPO group (77.2%). (**C**) The relationship between EPO doses and VDI was also demonstrated—the higher the EPO doses, the lower the VDI. *, *p* < 0.0001.

**Figure 13 cells-10-03133-f013:**
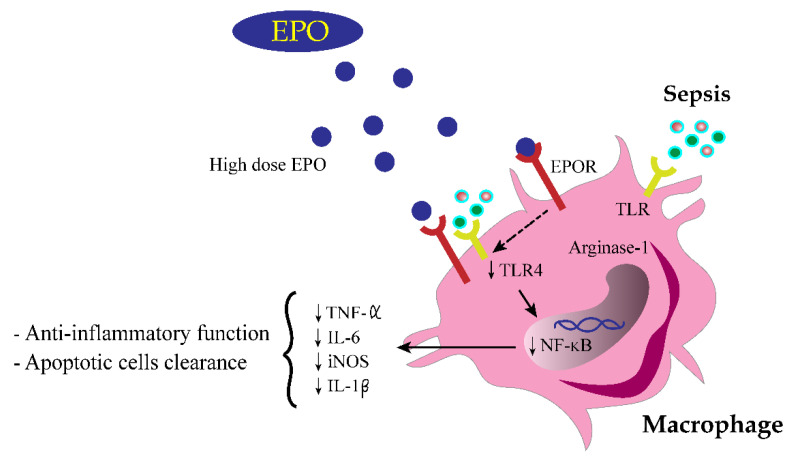
Schematic of the proposed hypothesis of sepsis attenuation by erythropoietin (EPO). DAMPs, damage-associated molecular patterns; IL, interleukin; EPO, erythropoietin; EPOR, erythropoietin receptor; iNOS, inducible nitric oxide synthase; NF-ΚB, nuclear factor-kappa B; PAMPs, pathogen-associated molecular patterns; TLR, toll-like receptor; TNF-α, tumor necrosis factor-alpha.

**Table 1 cells-10-03133-t001:** List of the primers used in this study.

Primers		
Inducible nitric oxide synthase (*iNOS*)	Forward	5′-CCCTTCCGAAGTTTCTGGCAGCAGC-3′
	Reverse	5′-GGCTGTCAGAGCCTCGTGGCTTTG-3′
Interleukin-1β (*IL-1β*)	Forward	5′-GAAATGCCACCTTTTGACAGTG-3′
	Reverse	5′-TGGATGCTCTCATCAGGACAG-3′
Arginase-1 (*Arg-1*)	Forward	5′-CAGAAGAATG GAAGAGTCAG-3′
	Reverse	5′-CAGATATGCA GGGA GTCACC-3′
Resistin-like molecule alpha (*Fizz-1*)	Forward	5′-GCCAGGTCCTGGAACCTTTC-3′
	Reverse	5′-GGAGCAGGGAGATGCAGATGAG-3′
Nuclear factor kappa B RelA (*NF**-**κB*)	Forward	5′-CTTCCTCAGCCATGGTACCTCT-3′
	Reverse	5′-CAAGTCTTCATCAGCATCAAACTG-3′
Toll-like receptor-4 (*TLR-4*)	Forward	5′-GGCAGCAGGTGGAATTGTAT-3′
	Reverse	5′-AGGCCCCAGAGTTTTGTTCT-3′
Erythropoietin receptor (*EPOR*)	Forward	5′-GTCCTCATCTCGCTGTTGCT-3′
	Reverse	5′-CAGGCCAGATCTTCTGCTG-3′
Signal transducer and activator of transcription 3 (*Stat3*)	Forward	5′-CTTGTCTACCTCTACCCCGACAT-3′
	Reverse	3′-GATCCATGTCAAACGTGAGCG-5′
Beta-actin (*β-actin*)	Forward	5′-CGGTTCCGATGCCCTGAGGCTCTT-3′
	Reverse	5′-CGTCACACTTCATGATGGAATTGA-3′

**Table 2 cells-10-03133-t002:** Baseline characteristics.

Characteristic ^1^	EPO Group (n = 151)	Non-EPO Group (n = 193)	*p*-Value
Age—yr	65.9 ± 13.8	57.1 ± 8.5	<0.0001
Sex—no. (%)			
Female	102 (67.5)	115 (59.6)	0.14
Sepsis—no. (%)	106 (70.2)	129 (66.8)	0.56
Septic shock—no. (%)	45 (29.8)	64 (33.2)	0.56
APACHE II score	21.9 ± 8.9	22.3 ± 8.1	0.66
Lactate level (mmol/L)	4.1 ± 2.0	3.9 ± 2.3	0.85
Vasopressor dependency index (VDI)	0.26 ± 0.2	0.30 ± 0.2	0.07
Specific diagnosis of sepsis on admission—no. (%)			
Respiratory disease	42 (27.8)	72 (37.3)	0.07
Genitourinary disease	94 (62.3)	118 (61.1)	0.91
Hepatobiliary and intestinal disease	15 (9.9)	3 (1.6)	0.0009
Medical history—no. (%)			
≥1 coexisting condition	151 (100)	114 (59.1)	<0.0001
Autoimmune disease	7 (4.6)	0 (0)	0.003
Cardiac disease	84 (55.6)	70 (36.3)	0.0005
Cerebrovascular disease	26 (17.2)	15 (7.8)	0.01
Chronic pulmonary disease	11 (7.3)	29 (15.0)	0.03
Chronic kidney disease	151 (100)	64 (33.2)	<0.0001
Diabetes mellitus	139 (92.1)	75 (38.9)	<0.0001
Hyperlipidemia	107 (70.9)	82 (42.5)	<0.0001
Hypertension	151 (100)	176 (91.2)	<0.0001
Liver disease	8 (5.3)	14 (7.3)	0.51
Solid organ tumor (cured)	4 (2.6)	7 (3.6)	0.76
Laboratory values			
Hemoglobin—g/dL	9.4 ± 1.1	11.3 ± 1.2	<0.0001
Reticulocytes—%	2.0 ± 0.4	1.8 ± 0.6	0.0005
Iron—µg/dL	23.8 ± 20.5	26.9 ± 18.4	0.14
Ferritin—ng/mL	530.8 ± 256.2	569.1 ± 196.5	0.12
Transferrin saturation—%	16.7 ± 11.3	19.7 ± 18.2	0.08
Creatinine—mg/dL	3.6 ± 2.1	1.8 ± 1.3	<0.0001
eGFR (mL/min/1.73 m^2^)	26.4 ± 22.2	38.9 ± 26.7	<0.0001
EPO dose prior to ICU admission			
<8000 units/week	48 (31.8)	—	
8000–16,000 units/week	101 (66.9)	—	
>16,000 units/week	2 (1.3)	—	
Duration of therapy—months, median (IQR)	39.7 (28.7–45.1)	—	
Short-acting EPO—no. (%)	151 (100)	—	
Darbepoetin-α—no. (%)	—	—	
CERA—no. (%)	—	—	
Subcutaneous route—no. (%)	125 (82.8)	—	
Intravenous route—no. (%)	26 (17.2)	—	
Renal replacement therapy ^2^			
Intermittent hemodialysis—no. (%)	34 (22.5)	46 (23.8)	0.77
Sustained low-efficiency dialysis—no. (%)	18 (11.9)	16 (8.3)	0.27
Continuous renal replacement therapy—no. (%)	7 (4.6)	11 (5.7)	0.65
Thrombotic vascular events—no. (%)	2 (1.3)	5 (2.6)	0.40

^1^ There was a small number of missing values for some variables, including the APACHE II score, iron, ferritin and transferrin saturation; however, each variable was reported for over 75% of participants. ^2^ In case of more than one modality performed during the study period, only the most sophisticated one (continuous renal replacement therapy > sustained low-efficiency dialysis > intermittent hemodialysis) has been reported. APACHE, Acute Physiology and Chronic Health Evaluation (APACHE II) score; CERA, continuous erythropoietin receptor activator; eGFR, estimated glomerular filtration rate calculated by the CKD-EPI equation [26]; EPO, erythropoietin; ICU, intensive care unit; IQR, interquartile ranges. To convert values for iron to micromoles per liter, multiply by 0.179. To convert values for creatinine to micromoles per liter, multiply by 88.4.

**Table 3 cells-10-03133-t003:** Baseline characteristics in low-dose and high-dose participant subgroups.

Characteristics	Low-Dose EPO Group (n = 48)	High-Dose EPO Group (n = 101)	*p*-Value
Age—yr	63.4 ± 8.8	66.3 ± 10.2	0.26
Female gender—no. (%),	24 (50)	77 (76.2)	0.001
Sepsis—no. (%)	34 (70.8)	72 (71.3)	0.95
Septic shock—no. (%)	14 (29.2)	31 (30.7)	0.85
APACHE II score	20.5 ± 4.2	22.6 ± 5.1	0.14
Lactate level (mmol/L)	3.7 ± 2.4	4.3 ± 2.3	0.71
Vasopressor dependency index (VDI)	0.25 ± 0.2	0.30 ± 0.2	0.99
Specific diagnosis of sepsis on admission—no. (%)			
Respiratory disease	10 (20.8)	32 (31.7)	0.17
Genitourinary disease	33 (68.8)	61 (60.3)	0.32
Medical history—no. (%)			
≥1 coexisting condition	48 (100)	101 (100)	-
Cardiac disease	25 (52.1)	57 (56.4)	0.62
Cerebrovascular disease	10 (20.8)	15 (14.9)	0.01
Chronic kidney disease	48 (100)	101 (100)	-
Diabetes mellitus	43 (89.6)	94 (93.1)	0.46
Hypertension	48 (100)	101 (100)	-
Liver disease	2 (4.2)	6 (5.9)	0.67
Solid organ tumor (cured)	3 (6.3)	1 (1.0)	0.06
Laboratory values			
Hemoglobin—g/dL	9.4 ± 0.6	9.5 ± 0.5	0.13
Reticulocytes—%	1.7 ± 0.2	1.9 ± 0.2	0.99
Iron—µg/dL	22.1 ± 10.4	24.8 ± 11.4	0.49
Transferrin saturation—%	15.4 ± 10.7	18.3 ± 12.2	0.32
Creatinine—mg/dL	3.5 ± 1.2	3.8 ± 1.0	0.13
eGFR (mL/min/1.73 m^2^)	30.5 ± 11.4	25.7 ± 13.5	0.20
Route of EPO administration			
Subcutaneous route—no. (%)	37 (77.1)	86 (85.1)	0.23
Intravenous route—no. (%)	11 (22.9)	15 (14.9)	0.23
Renal replacement therapy ^1^			
Intermittent hemodialysis—no. (%)	15 (31.3)	19 (18.8)	0.09
Sustained low-efficiency dialysis—no. (%)	8 (16.7)	10 (9.9)	0.24
Continuous renal replacement therapy—no. (%)	3 (6.3)	4 (4.0)	0.54
Thrombotic vascular events—no. (%)	1 (2.1)	1 (1.0)	0.59

^1^ In case of more than one modality performed during the study period, only the most sophisticated one (continuous renal replacement therapy > sustained low-efficiency dialysis > intermittent hemodialysis) has been reported. APACHE, Acute Physiology and Chronic Health Evaluation (APACHE II) score; eGFR, estimated glomerular filtration rate calculated by the CKD-EPI equation; EPO, erythropoietin; ICU, intensive care unit.

**Table 4 cells-10-03133-t004:** Summary of data on red-cell transfusions, vasopressors and causes of death.

Variable	EPO Group (n = 151)	Non-EPO Group (n = 193)	Odds Ratio (95% CI)	*p*-Value
Patients receiving a transfusion—no. (%)	64 (42.4)	50 (25.9)	2.104 (1.333–3.320)	0.0018
Units transfused per patient	4.1 ± 3.8	3.6 ± 3.1		0.18
Changes in hemoglobin concentration from baseline to day 29	3.6 ± 2.2	4.8 ± 1.6		<0.0001
Vasopressor dependency index				
Day 1 (n = 45/151 vs. 64/193)	0.37 ± 0.2	0.42 ± 0.2		0.20
Day 2 (n = 41/151 vs. 62/193)	0.21 ± 0.2	0.28 ± 0.2		0.09
Day 3 (n = 40/151 vs. 60/193)	0.18 ± 0.2	0.26 ± 0.2		0.04
Day 4 (n = 40/151 vs. 60/193)	0.20 ± 0.2	0.32 ± 0.2		0.02
Day 5 (n = 40/151 vs. 60/193)	0.20 ± 0.1	0.25 ± 0.1		0.02
Causes of death				
Septic shock with multiple organ failure—no. (%)	12 (7.9)	31 (16.1)		0.02
Secondary acute coronary syndrome—no. (%)	3 (2.0)	1 (0.5)		0.20
Severe gastrointestinal bleeding—no. (%)	7 (4.6)	12 (6.2)		0.52

Plus–minus (±) values are the mean ± SD.

**Table 5 cells-10-03133-t005:** Cox proportional hazards regression analysis for sepsis patients with or without erythropoietin (EPO).

Variables	HR (95% CI) EPO Group (n = 151)	HR (95% CI) Non-EPO Group (n = 193)
Gender (male)	1.13 (1.02–1.18)	1.00 (0.97–1.02)
Age (65–69 vs. <65 years)	1.79 (1.45–1.89)	1.75 (1.52–1.84)
Age (70–79 vs. <65 years)	2.17 (2.02–2.28)	1.88 (1.80–1.95)
Diabetes mellitus (yes vs. no)	1.17 (1.10–1.20)	1.10 (1.06–1.15)
Hypertension (yes vs. no)	1.02 (0.97–1.05)	0.96 (0.90–1.02)
Previous cardiac disease (yes vs. no)	1.00 (0.96–1.04)	0.70 (0.66–0.72)
Previous cerebrovascular accident (yes vs. no)	0.99 (0.94–1.03)	0.62 (0.60–0.65)
Renal function (eGFR 15–29 vs. >60 mL/min/1.73 m^2^)	1.62 (1.34–1.70)	2.01 (1.94–2.11)
Renal function (eGFR < 15 vs. >60 mL/min/1.73 m^2^)	1.18 (1.14–1.22)	—

eGFR, estimated glomerular filtration rate.

## Data Availability

The data are available from the corresponding author upon reasonable request.

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
