# Peer review of "Repurposing of High-Dose Erythropoietin as a Potential Drug Attenuates Sepsis in Preconditioning Renal Injury"

_cells, 2021, doi:10.3390/cells10113133_

Round 1
Reviewer 1 Report
Chancharoenthana W. et al. reported the administration of high-dose erythropoietin as preconditioning against renal injury by sepsis. The results are clear but the authors used mouse model of sepsis, the murine macrophage cell line, the hepatocyte cell line and patients with sepsis. The stories are very complicated and difficult to understand. It is clear that high-dose Epo attenuates sepsis-induced renal injury, but the mechanisms proposed by the authors can not be confirmed by the data. EPOR mRNA expression in mice were shown but Western blot analysis of EPOR was shown by the cell lines. Only EPOR mRNA expressions were examined in organs of mouse model of sepsis. Blood pressure, BUN, creatinine, serum TNFα, IL6 and IL10 are the data from sepsis mice. It is concluded that high-dose Epo reduces inflammation and renal injury. The data obtained from macrophage are interesting but it is not known whether such data explain the mechanisms of septic mice. Although macrophages are known to be target cells of non-hematopoieic biological effects of Epo, the results obtained from macrophages cannot be applied to the mouse model of sepsis. The Western blot of NFκB and EPOR should be performed in mouse model of sepsis.
The authors showed that Epo attenuates spleen apoptosis in mouse model of sepsis. The qualities of these Figure 6B are poor. The administration of Epo may induce autophagy. The occurrence of autophagy is known to reduce the progression of acute kidney injury by ischemic reperfusion injury. How the authors consider this possibility?
The qualities of Fig. 2H are poor.
The Western blot analyses of NFkB or EPOR and beta-actin were shown in one gel. In the reference 18 (reference of method of Western blot), 24p3R and GAPDH were shown in different gels, suggesting the methods in the present study and the previous one are different. The authors should describe the methods accurately.
Reviewer 2 Report
The authors performed a study suggesting that short-term administration of high-dose EPO may have a favorable effect on survival of patients with AKI. The description is coordinate and reasonable. However, there are some concerns that need to be addressed.
1. A clear description of when mice were sacrificed in the method is required. For example, Author described that all mice administered CLP were sacrificed and analyzed after 96 hours. However, in figure 3, after bilateral nephrectomy, all mice died within 72 hours, and the 5/6 nephrectomy group died within 48 hours. It is unclear whether the days on which mice were sacrificed were different. It is necessary to divide all experimental groups into each, and describe when the operation was performed and when the sacrifice was made.
2. In all western blots, NFkB/pNFkB/EPOR and actin are thought to be the result of attaching primary antibodies together on the same gel. However, although there is little possibility, the cross-reaction of each primary antibody is concerned. it can nonspecifically attach to bands of different sizes, so they can interfere with each other.
Is there any result of attaching the target protein and beta actin, respectively, by cutting the gel from about 50 kd size line?
3. In human study, the results are very interesting. In the group treated with EPO, they were older, had more severe anemia, had a lower renal function, and had many underlying diseases including diabetes. However, the patients with just short duration EPO Tx had better survival compared to non EPO treated patients. It is not easy to understand that EPO alone overcame these disadvantages and resulted in better 29-day survival. Could you present the number and duration of patients who underwent renal replacement therapy such as CRRT? Also, Could you present the cause of death for each group?
In addition, it would be better if it showed the hazard ratio that the administration of EPO reduced the risks for each factor such as age (over 65/under), sex, race, renal function, blood pressure, and presence or absence of diabetes.
4. In the inclusion criteria, the patient was already received EPO for an underlying disease. If so, were all the patients received the short-acitng EPO? were some patients received CERA or darpeboietin? what was the total duration of EPO administration?
5. In the case of a solid tumor, if it is an active tumor, EPO administration is generally reluctant, but was it not in an active state? In the case of tumor, was it a patient who had been cured?
Reviewer 3 Report
It was a great work. I have ten questions or comments and you can find attached file.

Round 2
Reviewer 1 Report
The manuscript improved very much. I have no further comments.
Author Response
We would like to express our greatest appreciations for the acceptance of our manuscript from the reviewer.
Reviewer 2 Report
The authors addressed most of my suggested corrections and concerns. the revised version of the manuscript appears to be good
Author Response

(The authors gave the same response as above.)

Reviewer 3 Report
1. On 3.1. Sepsis Enhanced Endogenous EPO Production and Reduced EPO Receptor in the Liver but Was Attenuated by High-Dose EPO.
-> This chapter has large contents. How about divide two part?
1) Sepsis enhanced endogenous EPO production with or without renal injury model, and reduced EPO receptor expression in the liver.
2) Mortality, pre-conditioning renal injury, and inflammaiton in sepsis were attenuated by high-dose EPO
2. Some data was changed, would you like to refine the abstract?
3. In the "Characteristics" column of Table 3, delete all total numbers of each RRT.
